# Urinary titin is not an early biomarker of skeletal muscle atrophy induced by muscle denervation in mice

**Jun Tanihata** [ID]*, **Susumu Minamisawa**

Department of Cell Physiology, The Jikei University School of Medicine, Nishishinbashi, Minato-ku, Tokyo, Japan

* tanihata@jikei.ac.jp

## Abstract

Early detection of skeletal muscle atrophy is important to prevent further muscle weakness. However, there are few non-invasive biomarkers for skeletal muscle atrophy. Recent studies have reported that the N-terminal fragment (N-titin) of titin, a giant sarcomeric protein, is detected in the urine of patients with muscle damage. In this study, we hypothesized that urinary N-titin would be a potential early biomarker of skeletal muscle atrophy in mice caused by sciatic nerve denervation. Male mice were randomly divided into control and denervation groups, and urinary N-titin levels were assessed daily for 9 days using an enzyme-linked immunosorbent assay system. Despite reduced titin protein levels in atrophic muscles 10 days after denervation, cleaved N-titin fragments were not increased in the urine of mice with denervation-induced muscle atrophy. Furthermore, we found no uptake of Evans blue dye from the extracellular space into the cytoplasm in atrophic muscles, suggesting that the sarcomeric membrane is intact in those muscles. The present results suggest that cleaved N-titin in the urine is not suitable as an early biomarker of skeletal muscle atrophy.

**Data Availability Statement:** All relevant data are within the manuscript and its Supporting Information files.

## Introduction

Skeletal muscle atrophy due to microgravity [1], disuse [2], cancer [3], and aging [3] directly leads to muscle weakness and is strongly associated with reduced quality of life [4]. Therefore, the early detection of muscle atrophy is important to its prevention and reduction. Imaging methods such as dual-energy X-ray absorption measurement (DXA) and computed tomography (CT), walking speed tests, and 6-min walking tests are used to detect muscular atrophy [5]. However, defining muscle atrophy using the above-mentioned tests is not always easy. A simple and reliable screening tool for the diagnosis of muscle atrophy is very important and needed [6]. Biomarkers could serve as objective, early indicators of the biological process of muscle atrophy [7]. Several studies have suggested that inflammatory biomarkers such as C-reactive protein, interleukin-6, tumor necrosis factor-$\alpha$ [8], and irisin [9] could be useful for the detection of muscle atrophy due to aging. Recent studies have also demonstrated that the Wnt antagonist frizzled-related protein (FRZB) is a biomarker for denervation-induced

**Funding:** This research was supported by the Jikei University Research Fund (SM), the Descente and Ishimoto Memorial Foundation of the Promotion of Sports Science (JT), and JSPS KAKENHI (Grant Number 16K08726) (JT).

**Competing interests:** The authors have declared that no competing interests exist.

muscle atrophy in amyotrophic lateral sclerosis (ALS) [10] and creatinine in spinal muscular atrophy (SMA) [11]. However, these biomarkers have not been widely applied to diagnose muscle atrophy, because they were not specific to muscle atrophy, showed weak association with clinical outcomes, and were invasive due to the required blood sampling from patients [12]. Therefore, it is important to identify non-invasive and specific biomarkers that can detect the early stage of muscle atrophy.

Titin is a giant sarcomeric protein that is involved in muscular passive tension and visco-elasticity. According to previous reports [13], titin is one of the earlier myofibril proteins whose level is decreased by muscle atrophy. Importantly, the N-terminal titin fragment (N-titin) in urine was recently identified as a useful non-invasive biomarker for muscle damage such as that seen in Duchenne muscular dystrophy (DMD) [14, 15]. Urinary N-titin was also demonstrated to be increased in muscle damage due to non-alcoholic fatty liver disease [16] and in healthy volunteers engaged in endurance exercise [17, 18]by using the enzyme-linked immunosorbent assay (ELISA) system [14]. Furthermore, we found that urinary N-titin is increased in patients after cardiac surgery [19]. Nakanishi et al. recently reported that urinary N-titin levels were increased 10- to 30-fold in non-surgical adult patients staying in the intensive care unit (ICU) compared with normal levels, and that they were correlated with decreases in patient muscle mass [20]. According to Nakanishi et al.'s report [20], urinary N-titin could be a potential candidate as an early biomarker of skeletal muscle atrophy. However, there are no reports directly investigating whether urinary N-titin is increased with muscle atrophy using highly sensitive ELISA system. In the present study, we hypothesized that urinary N-titin would be a good indicator for detecting muscle atrophy in the early stage. To evaluate our hypothesis, urinary N-titin was measured in mice that underwent denervation of the sciatic nerve.

## Materials and methods

### Animals

Experiments were performed after obtaining approval from the Animal Experiment Committee of the Jikei University School of Medicine. Male C57BL/6J mice were purchased from Nihon CREA. Mice were allowed free access to a pelleted laboratory animal diet and tap water.

Experiment 1: At 10 weeks of age, mice were randomly divided into two groups: a control (CON) group and a denervated (DEN) group (n = 10 each group). Mice in the DEN group underwent sciatic nerve transection surgery as previously described [21]. Briefly, mice were anesthetized using isoflurane, and a small incision was made in the posterior aspect of both hindlimbs to expose the sciatic nerve at the level of the femoral trochanter. At least 5.0 mm of the sciatic nerve was excised using small operating scissors. The skin was then closed with surgical glue. Mice in the CON group were also anesthetized using isoflurane and a small incision was made in the posterior aspect of both hindlimbs, which was then closed.

Experiment 2: At 10 weeks of age, mice were randomly divided into two groups: a control (CON) group and a dexamethasone-administered (DEX) group (n = 5 each group). Briefly, dexamethasone (dose = 10mg/kg body weight) was intraperitoneally administered to mice for 9 consecutive days during experimental periods. In the CON group mice, an equivalent volume of 5% DMSO and 1% Tween 80 dissolved in saline was administered in the same manner.

### Urine sample collection and measurement of N-titin

Mice were individually housed in metabolic cages during the experimental period.

Experiment 1: For urine samples, pooled urine was used every 24 h pre-operatively and post-operatively from day 1 to day 9. Collected urine samples were stored at -20˚C until analysis.

Experiment 2: For urine sample, pooled urine was used before administration and every 24 h administration period from day 1 to day 9 without day 4 and day 7.

Collected urine samples were stored at -20˚C until analysis. The urinary levels of N-titin were measured using the ELISA system (#27602 Mouse Titin N-fragment Assay Kit, Immuno-Biological Laboratories) according to the manufacturer's instructions. To avoid the effects of urinary filtrations, the value of N-titin concentrations was corrected by the value of creatinine and is shown by the following creatinine ratio: (N-titin/Cr) = N-titin (pmol/L)/creatinine (mg/dL). Urine creatinine concentration measurement was carried out with Lab Assay™ Creatinine (Wako Pure Chemical Industries). The samples were analyzed in duplicate and averaged for each measurement.

## Measurement of serum creatine kinase (CK)-MB levels

For obtain of serum, blood samples were collected from the abdominal aorta at the sacrificed and the blood was centrifuged at 3000 rpm, 15 min to obtain serum. For CK-MB was measured used by Fuji Dri Chem NX-500 (Fujifilm).

## Tissue preparation

Mice were sacrificed by cervical dislocation on post-operative day 10 or on administered day 9. The body and wet muscles were weighed. The tibialis anterior (TA) and soleus (SOL) muscles were collected using standard dissection methods. These muscles were frozen in liquid nitrogen for RNA and protein analysis.

## Real-time polymerase chain reaction (PCR) analysis

RNA isolation from the TA and SOL muscles in the CON and DEN groups and cDNA synthesis were performed as described [22]. Atrogin-1, Muscle RING Finger-1 (MuRF1), and titin expression levels were measured by real-time polymerase chain reaction (PCR) using the SYBR Premix Ex TaqII (TAKARA). The expression levels of these genes were normalized to those of 18S rRNA.

## Detection of titin by electrophoresis

Sodium dodecyl-sulfate polyacrylamide gel electrophoresis (SDS-PAGE) for the detection of titin was conducted as per a previous study [23]. Protein samples of the TA and SOL muscles in the CON and DEN groups were prepared by Laemmli's buffer system [23]. The preparations were then solubilized and electrophoresed on 2% polyacrylamide slab gel that included 0.5% agarose. Finally, the gels were stained with Coomassie brilliant blue (CBB). These gels were created as per Tatsumi and Hattori's method [24]. The quantification of titin levels was carried out by the National Institutes of Health Image J software.

## Western blot analysis

Protein was extracted from the TA and SOL muscles in the CON and DEN groups. Western blot analysis was performed as previously described [25]. The primary antibodies used in this study were as follows: anti-connexin 43 (abcam) and anti-GAPDH (Cell Signaling Technology). The band intensities of the target proteins were analyzed using the National Institutes of Health Image J software and normalized by the band intensity of GAPDH.

## Myofiber damage evaluation

For Evans blue dye (EBD)-injected mice, animals were interperitoneally injected with a 1% EBD solution at a dose of 50 mg/kg on post-operative day 5 and sacrificed by cervical dislocation 24 h after EBD injection. The TA and SOL muscles from the CON and DEN groups were sampled and then frozen in isopentane cooled by liquid nitrogen for myofiber damage evaluation. Cryosections of 8 μm thickness were cut from these muscles. Each cryosection was observed under a fluorescence microscope at 10× magnification using BZ-9000 (KEYENCE) to estimate the area of myofiber.

## Immunohistochemistry

Cryosections of 8 μm thickness were cut from the TA and SOL muscles. Immunohistochemistry was performed as described previously [26]. Sections were stained with polyclonal rabbit antibodies directed against connexin 43 (abcam), and Alexa 488-conjugated goat anti-rabbit IgG antibody (Invitrogen) was used as a secondary antibody. Fluorescence images were obtained using a BZ-9000 fluorescence microscope (KEYENCE).

## Statistical analysis

All data are presented as means ± standard error of the mean (SEM). Statistical differences were assessed by an unpaired $t$-test or one-way repeated measures analysis of variance (ANOVA). A $p$ value < 0.05 was considered statistically significant.

## Results and discussion

### Titin was decreased at the protein level in muscles with denervation-induced atrophy

Ten days after sciatic nerve denervation, the weights of the TA and SOL muscles were 0.68- and 0.62-fold smaller in the DEN group compared with those in the CON group, respectively (Fig 1A), as previously reported [27]. The body weights were not different between the CON (24.8 ± 0.5 g) and DEN groups (25.0 ± 0.4 g). According to previous reports [13], titin is one of the earlier myofibril proteins whose level is decreased by sciatic nerve denervation. Therefore, we examined the expression levels of titin mRNA and protein in the TA and SOL muscles. Although the titin expression of the TA and SOL muscles was not significantly different at the mRNA level between the two groups (Fig 1B), the titin protein levels in the TA and SOL muscles were significantly decreased in the DEN group (Fig 1C and 1D). We have observed that titin in the gastrocnemius muscle is also reduced by denervation (S1 Fig). This result indicated that the downregulation of titin in muscle atrophy after denervation occurred at the post-transcriptional level, which is consistent with the results of previous studies [13]. Because titin protein is known to be degraded by E3-ubiquitin ligase such as MuRF1, we next examined the mRNA expression levels of muscle-specific E3-ubiquitin ligases. Quantitative PCR analysis revealed that MuRF1 and atrogin-1 mRNAs were significantly increased in the denervated TA and SOL muscles (Fig 2A and 2B). Although we have not examined the MuRF1 protein level, MuRF1 is known to be upregulated [27] and its activity is also enhanced [13] from the early stage of denervation. Therefore, it is suggested that the expression level of titin in protein is decreased due to protein degradation through E3-ubiquitin ligase such as MuRF1 from the early stage after denervation.

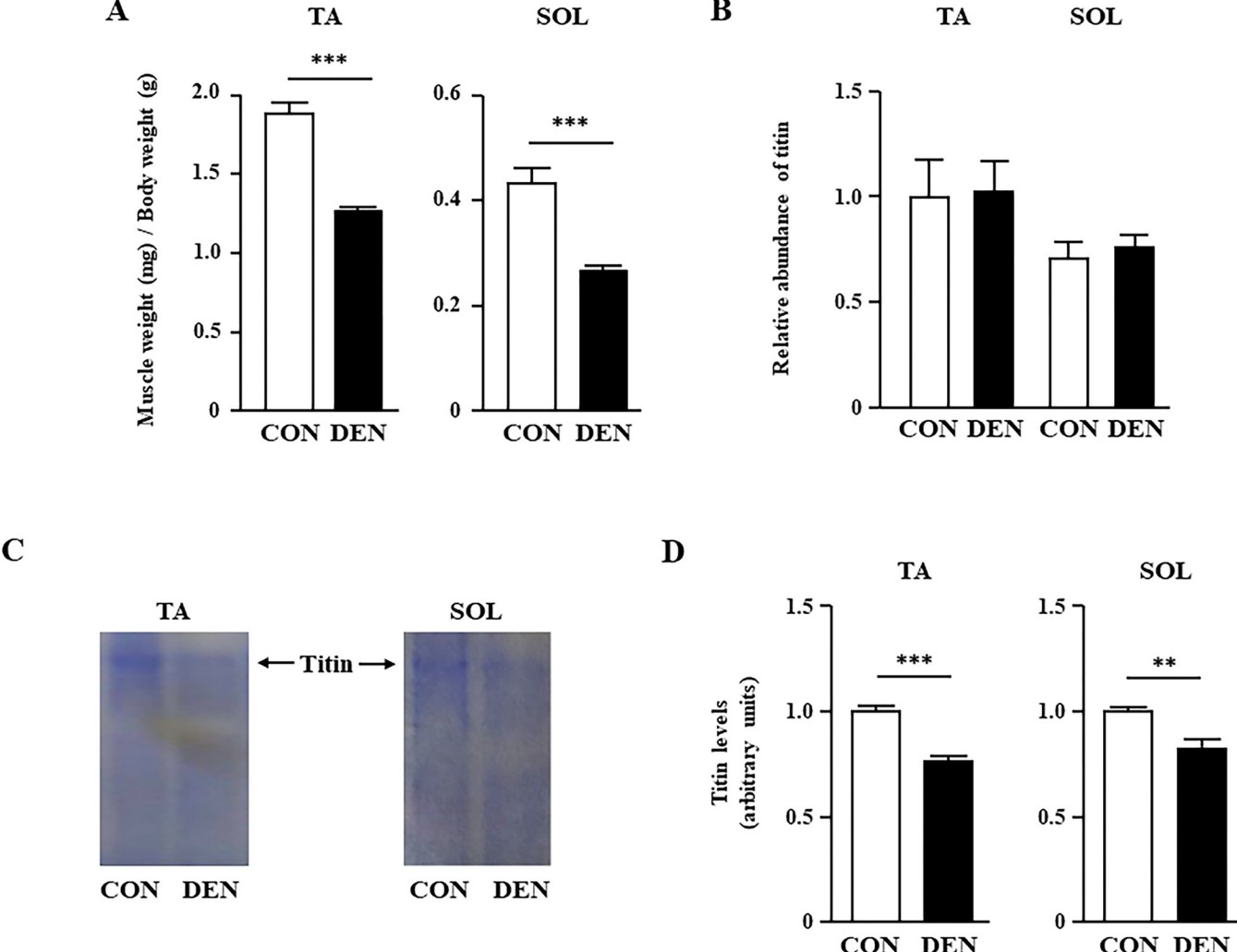

**Fig 1. The effects of muscle atrophy induced by denervation on muscle weight and titin expression levels.** (A) The weight in the TA and SOL muscles of the CON and DEN groups on post-operative day 10. (B) Quantification of real-time PCR products for titin expression in the TA and SOL muscles of the CON and DEN groups on post-operative day 10. (C) Typical SDS-PAGE image showing titin level in the TA and SOL muscles of the CON and DEN groups. (D) Quantification of titin in the TA and SOL muscles of the CON and DEN groups on post-operative day 10. Data are presented as the means ± standard error of the mean (SEM). **$p < 0.01$ and ***$p < 0.001$ by an unpaired $t$-test.

### Muscle atrophy did not increase urinary N-titin levels after sciatic nerve denervation

Titin protein is also known to be cleaved by calpain and matrix-metalloproteinase-2 when protease activity is increased in pathological conditions such as oxidative stress [28, 29]. Previous studies have demonstrated that cleaved titin fragments can be detected in the striated muscle that is associated with muscle damage in urine [14, 15, 18]. Furthermore, a recent study also demonstrated that the urinary titin level increased 10- to 30-fold compared with the normal level in nonsurgical, critically ill patients [20]. The authors suggested that the increased urinary titin levels reflected lower limb muscle atrophy. Therefore, we examined whether sciatic nerve denervation–induced muscle atrophy is associated with the increase in cleaved titin fragments in urine. In the present study however, we found that urinary N-titin/Cr levels were not different at any time point 10 days after the operation (Fig 3A), which is not consistent with the

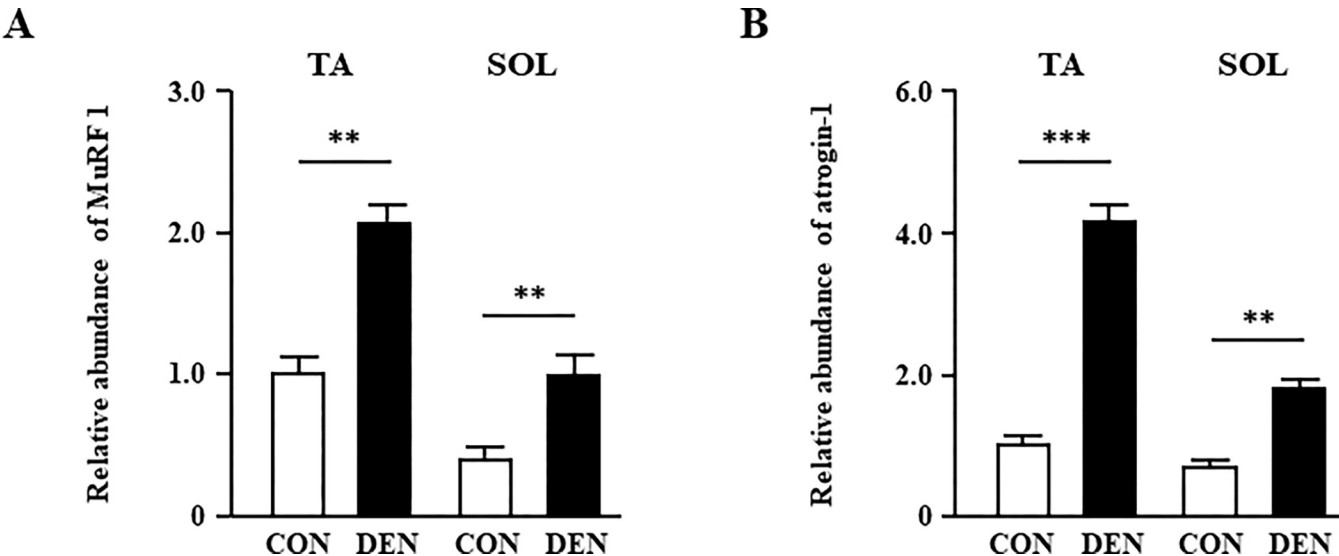

**Fig 2. The effects of muscle atrophy induced by denervation on the mRNA expression levels of muscle atrophy–related gene.** Quantification of real-time polymerase chain reaction (PCR) products for MuRF1 (A), and atrogin-1 (B) expression in the TA and SOL muscles of the CON and DEN groups on post-operative day 10. Data are presented as means ± standard error of the mean (SEM). **$p < 0.01$ and ***$p < 0.001$ by an unpaired $t$-test.

results of the previous study by Nakanishi et al. [20]. It should be noted that urinary N-titin/Cr levels were significantly increased in both CON and DEN groups on post-operative day 1 when compared to the level before operation, although there was no significant difference between DEN and CON groups (Fig 3A). Urinary N-titin/Cr levels were then rapidly decreased day by day and returned to the pre-operative levels around 7 days after the operation. By the 9th post-operative day, the levels of urinary N-titin were not different between the DEN and CON groups. Serum creatine kinase (CK)-MB, which is commonly used as a marker of muscle damage in muscle disease [26, 30] was elevated in both CON and DEN groups on post-operative day 1 compared to the pre-operation (S2A Fig) and then also returned to the pre-operative levels at 3 days after the operation (S2B Fig). In addition, muscle atrophy had not occurred in both CON and DEN groups on post-operative day 1 when compared to the levels before operation (S3 Fig). Therefore, we think that the transient increase in urinary N-titin/Cr levels immediately after the surgical operation could be due to muscle damage by surgical incision.

To confirm the effectiveness of the ELISA kit (#27602 Mouse Titin N-fragment Assay Kit; Immuno-Biological Laboratories) used in this experiment, we examined urinary N-titin/Cr in dystrophin and utrophin double-knockout (dKO) mice that exhibit severe muscular dystrophy compared to the most common murine DMD model, *mdx* mice [31]. We found that urinary N-titin/Cr levels in the dKO mice increased 15-fold compared with that in wild-type mice (Fig 3B). In contrast, the degree of N-titin/Cr increase was approximately 100-fold higher in DMD patients than in healthy volunteers according to the results of the ELISA kit used for human urinary titin measurement (#29501 Human Titin N-fragment Assay Kit, Immuno-Biological Laboratories) [14]. These results suggest that the sensitivity of the ELISA kit used in this experiment may be lower than that of the ELISA kit for measuring muscle damage in DMD patients [14] and ICU patients [20]. Next, we used another ELISA Kit (#SEB667 ELISA Kit for Titin, Cloud-Clone Corp.) provided by a different company to test the sensitivity for measuring mouse urinary titin using the same samples. We found that urinary N-titin/Cr levels in denervation-induced atrophy mice were very low, even at the first post-operative day and were not significantly different from those in wild-type mice (Fig 3C). Furthermore, the urinary N-

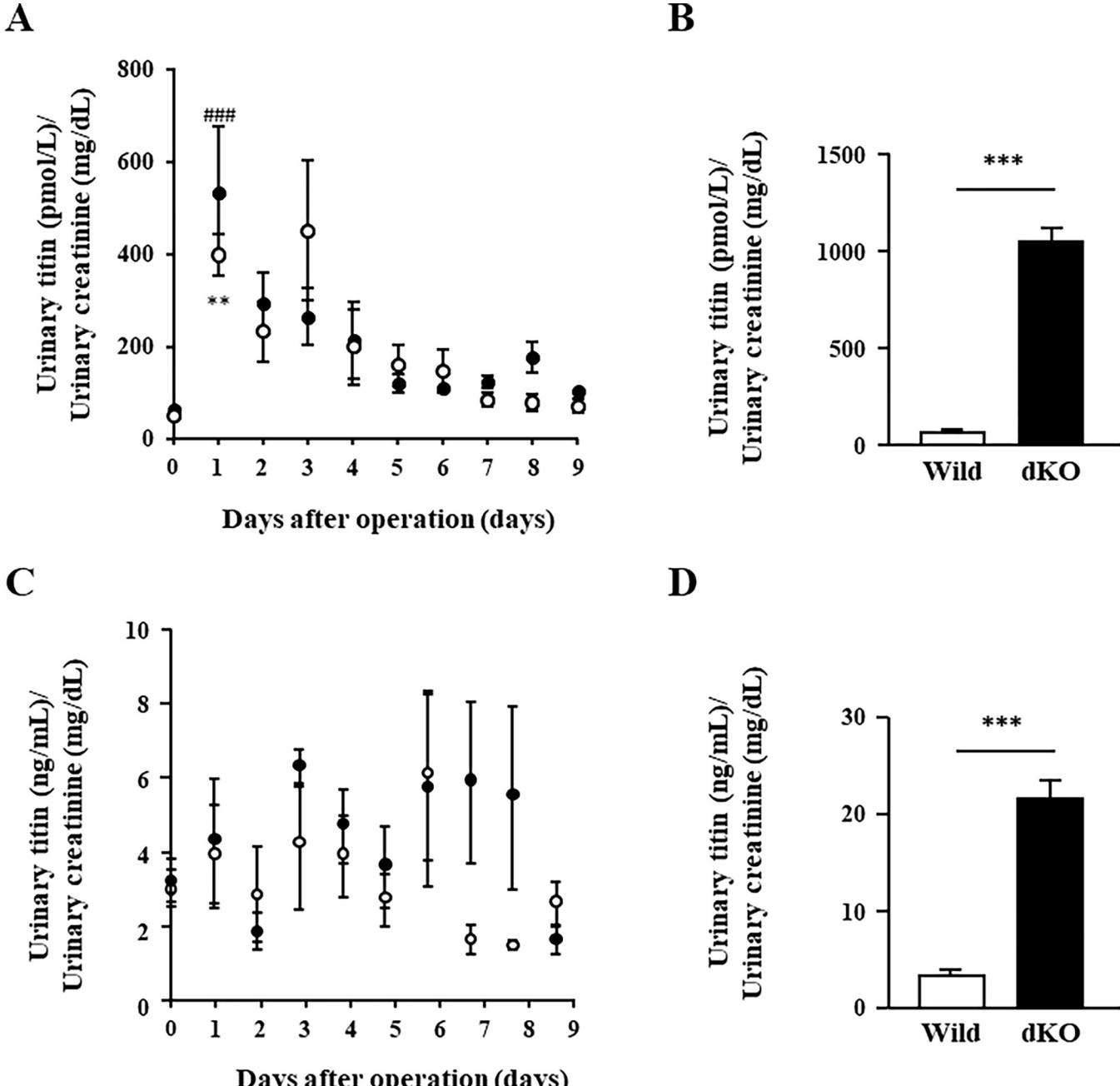

**Fig 3. The effects of muscle atrophy induced by denervation on urinary N-titin/Cr values.** (A) Time course changes in urinary N-titin/Cr levels used #27602 Mouse Titin N-fragment Assay Kit. Open circles: CON group. Closed circles: DEN group. Data are presented as means ± standard error of the mean (SEM). **$p < 0.01$ (0 days vs. 1 day after the operation in the CON group) and ### $< 0.001$ (0 days vs. 1 day after the operation in the DEN group) by one-way repeated measures analysis of variance (ANOVA). (B) Urinary N-titin/Cr values in severe muscular dystrophy models. Values of urinary N-titin/Cr in wild-type and dKO mice. Data are presented as means ± standard error of the mean (SEM). n = 3/group. ***$p < 0.001$ by an unpaired *t*-test. (C) Time course changes in urinary N-titin/Cr levels used #SEB667 ELISA Kit. Open circles: CON group. Closed circles: DEN group. Data are presented as means ± standard error of the mean (SEM). (D) Urinary N-titin/Cr values in severe muscular dystrophy models. Values of urinary N-titin/Cr in wild-type and dKO mice. Data are presented as means ± standard error of the mean (SEM). n = 3/group. ***$p < 0.001$ by an unpaired *t*-test.

titin/Cr levels in the dKO mice were increased only 6-fold compared with that in wild-type mice (Fig 3D), suggesting that the sensitivity of the #SEB667 ELISA Kit is much lower than that of the #27602 Mouse Titin N-fragment Assay Kit.

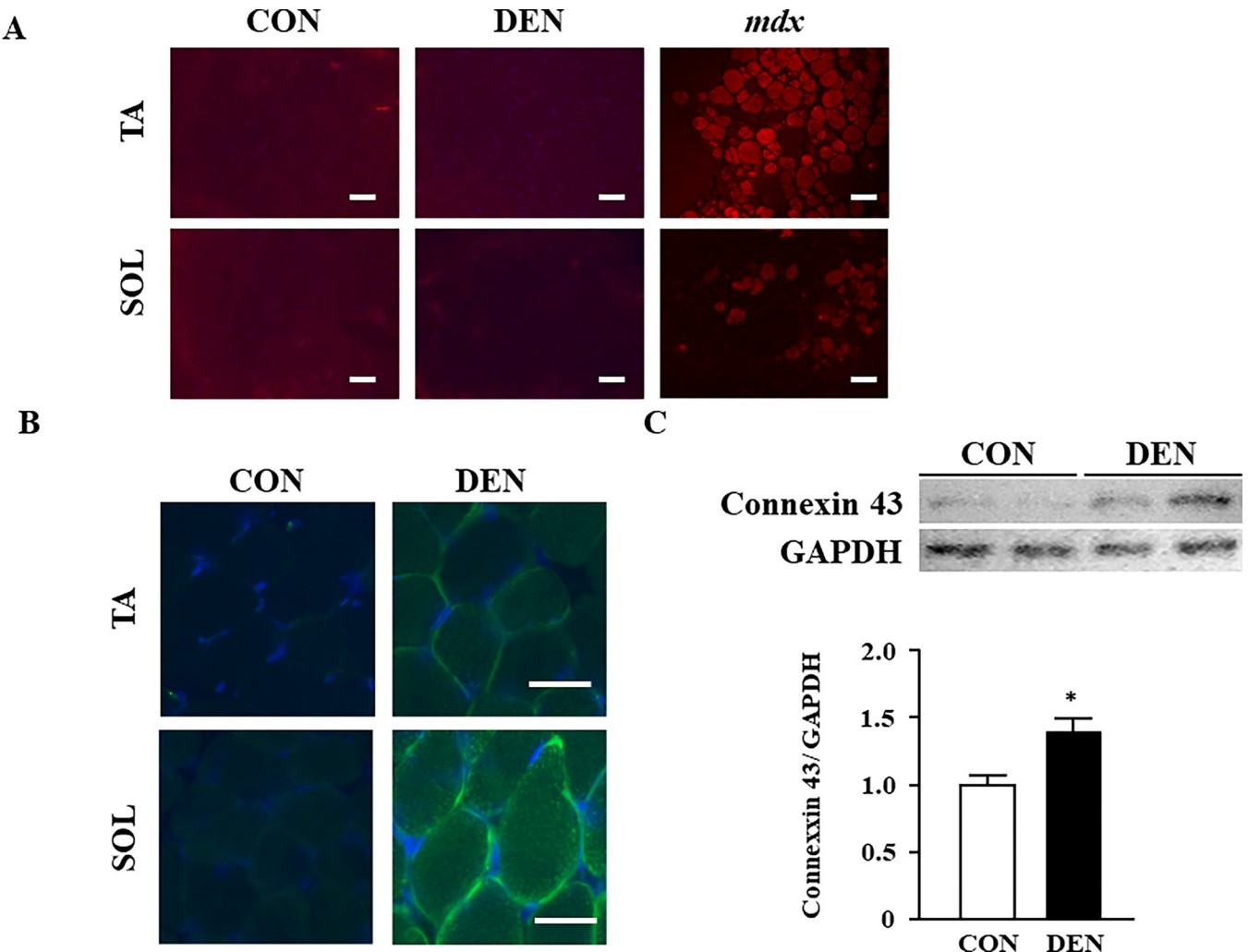

**Fig 4. The effects of muscle atrophy induced by denervation on the stabilization of muscle membrane and expression levels of connexin 43.** (A) Evans blue staining in the TA and SOL muscles of the CON and DEN groups on post-operative day 6. The TA muscle of *mdx* mice is shown as a positive control. Scale bar: 100 μm. (B) Immunohistochemical staining of connexin 43 in the TA and SOL muscles. Scale bar: 50 μm. (C) Western blot of connexin 43 in the TA muscle. The bar graph shows relative connexin levels compared to those of GAPDH. Data are presented as means ± standard error of the mean (SEM). n = 6/group. *$p < 0.05$ by an unpaired *t*-test.

Moreover, to further clarify the relationship between muscle atrophy and urinary titin levels, we examined urinary titin levels when muscle atrophy was induced by synthetic glucocorticoids, dexamethasone (DEX). The body weight in the DEX group was lower than that in the CON group at administration days 8 and 9 (S4A Fig). The weight of TA muscle was significantly decreased (S4B Fig), and although not shown in the data, the weights of gastrocnemius muscle, extensor digitorum longus muscle and plantaris muscles were also significantly decreased. On the other hand, the weight of SOL muscle did not decrease (S4B Fig). This finding that the effects of dexamethasone varied by muscle fiber type is consistent with previous reports [32]. Next, we also found that urinary N-titin/Cr levels were not different at any time point during administration periods (S5 Fig). Serum CK-MB levels were also unchanged on day 9 of DEX group compared to the CON group and pre-administration (S6 Fig). The results of no increase in serum CK-MB levels after induction of muscle atrophy by denervation or dexamethasone administration are consistent with previous reports [33].

## Sarcomeric membrane was intact in muscles with denervation-induced atrophy

To clarify the reason why fragmented titin did not leak into the urine due to denervation-induced muscle atrophy, the stabilization of the muscle membrane was examined using EBD. EBD uptake was not observed in the TA and SOL muscles in the CON and DEN groups, indicating that the sarcolemma was stabilized even after denervation-induced muscle atrophy (Fig 4A). Because previous studies have demonstrated that small molecules such as EBD are permeabilized via connexin 43 hemichannels on the sarcolemma of fast myofibers [34] and that the expression of connexin is increased in atrophic muscle induced by both denervation and dexamethasone [35], we analyzed the expression levels of connexin 43 in denervated muscle. The expression levels of connexin 43 in the TA and SOL muscles were significantly increased in the DEN groups (Fig 4B and 4C), which is consistent with the results of previous studies. Connexin hemichannels can release or take up only the monovalent cation, $Ca^{2+}$, and molecules with a molecular weight below 1.5 kDa [36]. Because the molecular weight of the N-titin fragment is approximately 21.4 kDa, the N-titin fragment cannot go through the connexin hemichannels even though the connexin 43 is upregulated in atrophic muscle. Therefore, we think that the N-titin fragment did not leak into the blood and urine to a detectable level even though it was degraded and fragmented by denervation-induced muscle atrophy. Another possibility is that the amount of titin released in the urine is small because muscle atrophy due to denervation occurs only in the hindlimb muscles whereas muscle damage in DMD patients occurs throughout the whole body. Our study also suggests that connexin 43 upregulation is not sufficient to induce EBD leakage from sarcolemma of denervation-induced atrophic muscles.

## Conclusion

We found that the cleaved N-titin fragment was not increased in the urine of mice with skeletal muscle atrophy 10 days after sciatic nerve denervation, even though the expression levels of titin protein were decreased. Therefore, the urinary N-titin fragment is not suitable for detecting the early stage of denervation-induced skeletal muscle atrophy.

## Supporting information

**S1 Fig. Typical SDS-PAGE image showing titin level in the gastrocnemius muscle of the CON and DEN groups.**
(TIF)

**S2 Fig. The effects of muscle atrophy induced by denervation on serum creatine kinase (CK)-MB.** (A) Serum CK-MB levels (U/L) in pre-operation and 1day after sham-operation and denervation. (B) Time course changes in serum CK-MB. Open circles: CON group. Closed circles: DEN group. Data are presented as the means ± standard error of the mean (SEM). n = 4/group. *$p < 0.05$ by an unpaired *t*-test (Compared to pre-operation (0 day)).
(TIF)

**S3 Fig. The effects of muscle atrophy induced by denervation on body weight and muscle weight at 1day after operation.** The body weight and TA and SOL muscles weight of the CON and DEN groups on pre- or post-operative day 1.
(TIF)

**S4 Fig. The effects of muscle atrophy induced by DEX administration on body weight and muscle weight.** (A) Time course changes in the body weight. Open circles: CON group. Closed

circles: DEN group. Data are presented as means ± standard error of the mean (SEM).
*$p < 0.05$ and $p < 0.01$ (compared to the 0 day) by one-way repeated measures analysis of variance (ANOVA). (B) The weight in the TA and SOL muscles of the CON and DEX groups on administration day 10.
(TIF)

**S5 Fig. The effects of muscle atrophy induced by DEX administration on urinary N-titin/Cr values.** Time course changes in urinary N-titin/Cr levels. Open circles: CON group. Closed circles: DEX group. Data are presented as means ± standard error of the mean (SEM).
(TIF)

**S6 Fig. The effects of muscle atrophy induced by DEX administration on serum creatine kinase (CK)-MB.** Serum CK-MB levels (U/L) in pre-administration, CON group and DEX group on day 9.
(TIF)

**S1 File.**
(PDF)

**S2 File.**
(PDF)

**S3 File.**
(PDF)

## Author Contributions

**Conceptualization:** Jun Tanihata, Susumu Minamisawa.

**Data curation:** Jun Tanihata.

**Formal analysis:** Jun Tanihata.

**Funding acquisition:** Jun Tanihata, Susumu Minamisawa.

**Investigation:** Jun Tanihata.

**Methodology:** Jun Tanihata.

**Project administration:** Jun Tanihata.

**Writing – original draft:** Jun Tanihata.

**Writing – review & editing:** Susumu Minamisawa.

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
