## [Decision Letter · Decision Letter 0]

9 May 2023

PONE-D-23-12044Urinary titin is not an early biomarker of skeletal muscle atrophy induced by muscle denervation in micePLOS ONE

Dear Dr. Tanihata,

Thank you for submitting your manuscript to PLOS ONE. After careful consideration, we feel that it has merit but does not fully meet PLOS ONE’s publication criteria as it currently stands. Therefore, we invite you to submit a revised version of the manuscript that addresses the points raised during the review process.

We look forward to receiving your revised manuscript.

Kind regards,

Shinsuke Yuasa

Academic Editor

PLOS ONE

Journal Requirements:

In your cover letter, please note whether your blot/gel image data are in Supporting Information or posted at a public data repository, provide the repository URL if relevant, and provide specific details as to which raw blot/gel images, if any, are not available. Email us at plosone@plos.org if you have any questions

Reviewers' comments:

Reviewer's Responses to Questions

**Comments to the Author**

1. Is the manuscript technically sound, and do the data support the conclusions?

Reviewer #1: Partly

Reviewer #2: Partly

Reviewer #3: Partly

2. Has the statistical analysis been performed appropriately and rigorously? 

Reviewer #1: Yes

Reviewer #2: Yes

Reviewer #3: Yes

3. Have the authors made all data underlying the findings in their manuscript fully available?

Reviewer #1: Yes

Reviewer #2: Yes

Reviewer #3: Yes

4. Is the manuscript presented in an intelligible fashion and written in standard English?

Reviewer #1: Yes

Reviewer #2: Yes

Reviewer #3: No

5. Review Comments to the Author

Reviewer #1: In this manuscript, Tanihata and Minamisawa tested if urinary N-titin could be a potential early biomarker of skeletal muscle atrophy induced by sciatic nerve denervation in adult mice. They found that urinary levels of cleaved N-titin fragments were unaltered under the atrophic conditions, although full-length titin protein levels were reduced in muscle tissues. They found that the sarcomeric membrane was not injured by the denervation, probably through the upregulation of connexin 43. They concluded that cleaved N-titin is unlikely to be released into extracellular spaces from muscle cells and unaltered in the urine in the denervation model, and thus urinary cleaved N-titin is an unsuitable early biomarker for muscle atrophy.

-Figure 1: Since gene and protein levels were normalized by the total amounts of mRNAs and proteins, respectively, the titin levels should be relatively decreased in the atrophic muscle tissues. The authors explained that decreased titin protein was due to protein degradation mediated by E3-ubiquitin ligases such as MuRF1. However, there is a possibility that reduced mRNA levels also contributed to the reduction of protein levels in the atrophic muscle. How do they exclude this possibility?

-Figure 3: the authors showed that urinary N-titin/Cr levels were markedly increased in both CON and DEN groups on postoperative day 1 compared to the non-operation groups. This could be due to the muscle damage by surgical incision. However, such upregulation is not observed in Supplemental Figure 2. The authors should explain this discrepancy.

Did the authors measure the serum creatine kinase levels? If it is unchanged in the denervation condition, they can conclude that muscle injury is not evident.

-Supplemental Figures 1 & 2: supplemental figures should be included into the main figures so that the readers can easily compare the sensitivities of ELISA kits for the detection of titin protein.

Reviewer #2: The manuscript describes that the N-terminal titin fragment was not increased in the urine despite reduction of the titin protein levels in atrophic muscles after sciatic nerve transection in mice. The authors concluded that the urinary titin fragment is not a suitable biomarker for detecting the early stage of skeletal muscle atrophy.

There are several concerns as below:

1) As authors describe, skeletal muscle atrophy is a phenomenon resulted from various causes including disuse, denervation, aging, cachexia, as well as muscle diseases such as muscular dystrophies. Therefore, the underlying mechanism is supposed to differ depending on its cause. It should be clearly stated that sciatic nerve transection is a model of denervation and that this study was conducted to examine the efficacy of the urinary N-terminal titin as a biomarker detecting denervation-induced atrophy. The term skeletal muscle atrophy is used loosely throughout and the discussion on interpretation of the results seems imprecise. For instance, they discussed comparison with nonsurgical, critically ill patients, but it seems that the underlying pathology is not denervation.

2) Does cleaved N-terminal titin occur in skeletal muscle? In Fig.1C western blot analysis did not clearly show the N-terminal titin fragment. Could titin's cleavage occur outside the skeletal muscle after leakage?

3) The authors showed upregulation of connexin 43 in denervated muscle. They speculate the N-titin fragment cannot go through the connexin hemichannels because of its molecular size. If it is the case, why EBD uptake was not observed?

To address the above-mentioned questions, it is necessary to analyze the status of titin in the circulating blood under skeletal muscle atrophy caused by different etiologies.

Reviewer #3: The study by Tanihata and Minamisawa examined the importance of titin in skeletal muscle atrophy during denervation. The results show that there is no increase in the expression of urinary/muscle titin during denervation. Therefore, the authors highlight that the titin levels was not reflected lower limb muscle atrophy. Unfortunately, most part of this work relies on muscle atrophy following denervation, an early atrophy model.

Me specific concerns are outlined below.

The authors should examine other early muscle atrophy models, such as disuse, and disease (or dexamethasone), because this manuscript focusing early muscle atrophy.

Since the greatest increase in the titin was observed 1 day after denervation, inflammation should be considered. What was the degree of muscle atrophy following 1 day?

How was titin levels in other hindlimb muscle?

Why the author did not measure connexin 43 in the SOL muscle?

Minor

The authors should describe "expression" only for genes, not proteins.

- Figures: Show marker position for each blot image.

6. PLOS authors have the option to publish the peer review history of their article (what does this mean?). If published, this will include your full peer review and any attached files.

Reviewer #1: No

Reviewer #2: No

Reviewer #3: No

---

## [Author Response · Author response to Decision Letter 0]

3 Jul 2023

We thank Reviewers and editor for his/her thorough review of the manuscript.　We have responded to each of Reviewers' criticisms and have modified the manuscript accordingly. It is our hope that the revised version is now deemed acceptable.

---

## [Decision Letter · Decision Letter 1]

13 Jul 2023

Urinary titin is not an early biomarker of skeletal muscle atrophy induced by muscle denervation in mice

PONE-D-23-12044R1

Dear Dr. Tanihata,

We’re pleased to inform you that your manuscript has been judged scientifically suitable for publication and will be formally accepted for publication once it meets all outstanding technical requirements.

Kind regards,

Shinsuke Yuasa

Academic Editor

PLOS ONE

Additional Editor Comments (optional):

Reviewers' comments:

Reviewer's Responses to Questions

**Comments to the Author**

1. If the authors have adequately addressed your comments raised in a previous round of review and you feel that this manuscript is now acceptable for publication, you may indicate that here to bypass the “Comments to the Author” section, enter your conflict of interest statement in the “Confidential to Editor” section, and submit your "Accept" recommendation.

Reviewer #1: All comments have been addressed

Reviewer #2: All comments have been addressed

Reviewer #3: (No Response)

2. Is the manuscript technically sound, and do the data support the conclusions?

Reviewer #1: Yes

Reviewer #2: Yes

Reviewer #3: Partly

3. Has the statistical analysis been performed appropriately and rigorously? 

Reviewer #1: Yes

Reviewer #2: Yes

Reviewer #3: Yes

4. Have the authors made all data underlying the findings in their manuscript fully available?

Reviewer #1: Yes

Reviewer #2: Yes

Reviewer #3: Yes

5. Is the manuscript presented in an intelligible fashion and written in standard English?

Reviewer #1: Yes

Reviewer #2: Yes

Reviewer #3: Yes

6. Review Comments to the Author

Reviewer #1: The revised manuscript has been improved. The authors have mostly addressed my concerns. I have no further comments.

Reviewer #2: According to my comments, the authors performed an additional experiment using a dexamethasone-induced muscle atrophy model and showed the consistent result that urinary titin levels did not change. Overall appropriate revisions has been made to the reviewer's concerns.

Reviewer #3: The authors have addressed most of my concerns. The more detailed methods and new results have significantly improved the manuscript. This reviewer has no further comments/critiques for this manuscript.

7. PLOS authors have the option to publish the peer review history of their article (what does this mean?). If published, this will include your full peer review and any attached files.

Reviewer #1: No

Reviewer #2: No

Reviewer #3: No

---

## [Editor Report · Acceptance letter]

2 Aug 2023

PONE-D-23-12044R1 

Urinary titin is not an early biomarker of skeletal muscle atrophy induced by muscle denervation in mice 

Dear Dr. Tanihata:

I'm pleased to inform you that your manuscript has been deemed suitable for publication in PLOS ONE. Congratulations! Your manuscript is now with our production department. 

Kind regards, 

on behalf of

Dr. Shinsuke Yuasa 

Academic Editor

PLOS ONE